# Study on the Method of Matched Splice Grafting for Melon Seedlings Based on Visual Image

Pengyun Xu [1], Tong Zhang [1], Liping Chen [2,3], Wenqian Huang [3] and Kai Jiang [2,3,*]

[1] College of Mechanical and Electrical Engineering, Hebei Agricultural University, Baoding 071001, China; jdxpy@hebau.edu.cn (P.X.); zhangtong90hou@163.com (T.Z.)

[2] Research Center of Information Technology, Beijing Academy of Agriculture and Forestry Sciences, Beijing 100097, China; chenlp@nercita.org.cn

[3] Research Center of Intelligent Equipment, Beijing Academy of Agriculture and Forestry Sciences, Beijing 100097, China; huangwq@nercita.org.cn

[*] Correspondence: jiangk@nercita.org.cn; Tel.: +86-10-5150-3504

**Abstract:** Due to the cutting mechanism of the existing grafting machine, it cannot adjust the cutting angle in real time, resulting in low fitting precision on the cutting surfaces between the rootstocks and scion seedlings and, thus, seriously affecting the survival rate and quality of the grafting seedlings. In this paper, a kind of splice grafting method based on visual image is proposed, aiming at maximizing the joint rate between cutting surfaces of rootstocks and scion seedlings and realizing precise cutting and grafting of grafting machine. After analysis, we determined that melon rootstock seedlings have a structure of pith cavity inside, and the solid structure from the top of the pith cavity to the left and right base points of a growing point forms the important area of a cutting surface. In order to obtain the geometric model of the cutting surfaces of the seedlings, a visual image analysis system was established to identify, analyze, and model the pith cavity structure inside the rootstock seedling, as well as the external morphological characteristics, and the ultimate cutting angle of the rootstock seedling and cutting surface parameters were determined. By measuring the length of minor axis of scion seedlings in order to achieve the maximum joint rate, the optimal cutting angle of the rootstocks and scion seedlings was determined. Then grafting and seedling cultivation tests were carried out. The test results showed that the range of ultimate cutting angle on rootstock seedlings (*Cucurbita moschata*) was $18.21 \pm 1.92°$; the cutting angles of the rootstock (*Cucurbita moschata*) and scion seedlings (watermelon) were $22°$ and $19.68°$, respectively; the cutting surface length of the two was 4.96 mm; and the cutting surface thickness of the rootstock was 0.13 mm, all of which could satisfy the technological requirements of the matched splice grafting of melons. The research results can serve as a reference for the design in vision-guided precision cutting and real-time grafting operation on grafting robots.

**Keywords:** visual image; melon seedlings; splice grafting; matched grafting; cutting model; rootstock pith cavity

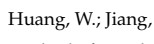



## 1. Introduction

Vegetable grafting can effectively overcome the soil-borne diseases and insect pests caused by continuous cropping [1]. After grafting, the ability of a plant root system to absorb fertilizers and water is significantly enhanced, and the yield can be increased by 20–50% [2,3]; therefore, grafting technology is extensively applied in the world [4,5]. In the last two decades, the vegetable seedling cultivation industry in China has enjoyed rapid growth. The demand for vegetable seedlings each year has exceeded 680 billion, and grafting technology has been applied in 90% of melons, but mostly relying on skillful workers [6–8]. Grafting is labor intensive. Generally speaking, grafting a huge number of seedlings is required to be finished in a short period of time; thus, manual grafting cannot adapt to large-scale and standardized production demands. Moreover, due to the high

indoor temperature and high humidity in the greenhouse, the workers operate while sitting on the seedling bed and endure extremely high labor intensity; thus, there are few workers willing to be engaged in this job [7,9]. With the population aging in rural China and the upsurge of manpower cost, seedling cultivation enterprises are in bad need of grafting machines; thus, the grafting machine has become the core equipment for industrialized seedling cultivation [10,11].

The technology of the grafting machine has always been the hot spot in agricultural robots in the world [12–14]. Melon grafting machine technology is popular in Asian countries, such as Japan, Korea, and China [15,16]; European countries, such as the Netherlands, Span, and Italy, are mainly focused on the research and development of grafting machine of solanaceous vegetables [17–20]. In 1994, ISEKI & Co. Ltd. in Japan worked with biological research institutes and launched a GR800B type semi-automatic grafting machine, which adopts the model of matched splice grafting for single seedlings and achieves production efficiency of 800 seedlings/h and success rate of 95% [21]. In 2011, ISEKI & Co. Ltd. (700 Umaki-cho, Matsuyama-shi, Ehime-ken, 799-2692 Japan) developed an automatic grafting machine which is equipped with an auxiliary automatic seedling feeding actuator to pick up seedlings from the seedling disk. The automatic seedling feeding actuator can adjust the direction of cotyledons in the process of seedling feeding and achieves seedling feeding rate over 90% [22]. The machine needs only one operator to supply seedling disk; therefore, the production efficiency of single-person operation achieves 800 seedlings/h, and other actuators have not been improved or upgraded [23]. In 2016, Helper Robotech Co. Ltd. (82, Yuha-ro 226beon-gil, Gimhae-si, Gyeongsangnam-do 621-834 Korea) in Korea launched the AFGR-800CS type vegetable grafting machine, which is used for grafting of melons and solanaceous vegetables; it achieves production a efficiency of 800 seedlings/h and a success rate of 95%. In order to solve the problem of cutting surface drift caused by the bending of seedling stem, a visual system was applied to identify and analyze the image information of the seedling cutting surface to determine the positional deviation of the cutting surface center of the seedling and center of the clamping hand. By adjusting the cutting surface centers of rootstocks and scion seedlings to make them fit with each other, the machine can further improve the grafting precision, but it cannot solve the problem of aligning the bent seedling stems [24,25].

The research and development units on grafting machine technology in China are mainly universities and research institutes [26]. Gu et al. (2005) designed a 2JC-350 automatic grafting machine with a cut grafting method for vegetable seedlings, requiring two operators to feed seedlings to achieve the production efficiency of 350 seedlings/h and success rate of 90%. It requires users to strictly control the diameter size of rootstocks and scion seedlings to ensure cut grafting operation. With low working efficiency and poor adaptability to seedlings, the machine is not acknowledged by customers nor extensively applied [27]. In 2010, the same research group worked with the Beijing Academy of Agriculture and Forestry Sciences and jointly developed a 2JC-600 oblique inserted grafting machine, which achieved production efficiency of 600 seedlings/h and a success rate of 93%; however, it failed in solving the flexible fitting between the insertion hole of the rootstock and size of the scion seedling stem [28,29]. Li et al. (2014) developed an oblique inserted grafting machine and designed an automatic rootstock clamping mechanism that can align seedlings and solved the problem of seedling damage during clamping due to bent seedling stems. They conducted an experimental study on the optimization of positioning parameters of rootstock cotyledons and determined the safety pressure for the adsorption of cotyledons, but the related research is still in the testing stage [30,31]. Wu et al. (2017) designed a grafting mechanism with the functions of automatic seedling feeding, grafting, transmission, and replanting; they also designed the root cutting and seedling feeding mechanism, seedling transmission mechanism, and replanting mechanism, which have improved automatic grafting, and the success rate in seedling feeding and transmission and replanting were 94% and 92%, respectively. However, related research is still in the testing stage [32,33]. In order to solve the problem of a long period of waiting in

manual seedling feeding, Jiang et al. (2012) developed a seedling transmission mechanism with a double-claw, which uses the servosystem to drive the transmission mechanism to do reciprocating and repeated operation. It has improved the positioning precision of the transmission mechanism and the operation efficiency of the machine, and the production efficiency can reach up to 848 seedlings/h, with a success rate of 95%; moreover, the research made a preliminary proposal and research method for the matched grafting of rootstocks and scion seedlings [34–37].

Wu et al. (2013) presented a machine vision system for restoring the cotyledons of seedlings and extracting their parameters by ellipse fitting. The test showed that 461 seedlings were identified and positioned in all 473 seedlings, and its rate reached 97.5%, which meets the requirement of robot grafting [38]. Wang et al. (2014) studied an automatic detection method for external features of grafting seedlings based on mathematical modeling. The detecting items included growth status, cotyledons parameters, hypocotyls parameters, and other external parameters [39]. However, there are few reports on the methods of seedling-matching grafting based on visual images.

In conclusion, current melon grafting machines have realized the standardized cutting of seedlings; however, the cutting components work according to fixed cutting angles, and the morphological differences between rootstocks and scion seedlings make it difficult to match each other, thus undermining the grafting precision and survival rate of grafted seedlings. For this reason, in this paper, a kind of melon splice grafting method based on visual image is proposed to acquire the image information of the inner and outer morphological characteristics of rootstocks and scion seedlings based on machine vision technology. By constructing the cutting model for matched grafting of rootstocks and scion seedlings, the working parameters, such as the cutting position and cutting angle, were determined to offer reference for the maximized fitting of cutting surfaces of grafted seedlings. The research results can offer a theoretical basis for the design in vision-guided precision cutting and real-time grafting operation on grafting robots.

## 2. Materials and Methods

### 2.1. Matched Grafting Method

In view of the difference in morphological characteristics between melon rootstocks and scion seedlings, in this paper, a matched grafting method is proposed to maximize their cutting surfaces. By constructing the geometric model of seedling cutting, the cutting angles of the rootstock–scion grafting were determined.

### 2.1.1. Grafting Method

Grafting methods for melons mainly include cut grafting, side grafting, and splice grafting [3]. Splice grafting is extensively applied in China. It has a high requirement for the matching degree of seedling age. If the scion seedling is too large in diameter, the rootstock seedling will be split up in cut grafting. If the scion seedling is too small in diameter, it will easily break off. In Shandong province in China, two proficient grafting workers can achieve a production efficiency of over 800 seedlings/h by cut grafting, but after the healing stage of the grafted seedling, it is necessary to eliminate sprout tillers manually, thus creating a very heavy workload [40]. Side grafting requires the planting of rootstock and scion seeds in the same feeding block at the same time to ensure the similar thickness of them. The stems of them undergo an up–down oblique cut to form a tongue-shaped cutting surface and then stick the two surfaces together and get fixed by a clip. After the survival of the seedlings, the root of the scion should be cut off. Although the survival rate is high by this method, the process is cumbersome, and it is not suitable for industrialized seedling cultivation [26]. Splice grafting requires users to cut one piece of cotyledon and the growing point and cut off the point 1–1.5 cm on the hypocotyledonary axis of the scion to get a cutting surface. Then they must fix the grafted seedling with a clip. Compared with cut grafting and side grafting, the cutting surface formed in splice grafting is flat and smooth for close fitting, thus achieving a higher survival rate of the grafted

seedlings [41]. It is more suitable for a mechanized grafting operation. Therefore, splice grafting is adopted on most current grafting machines for development and design. The aim of the splice grafting for melons proposed in this paper is to realize the same length of cutting surfaces of rootstocks and scion seedlings and improve the fitting precision and the survival rate of grafted seedlings.

### 2.1.2. Matching Model

There is a pith cavity structure in the rootstock seedlings of melons. The higher the seedling age, the larger the pith cavity; however, a large pith cavity is not suitable for grafting operations. If the pith cavity is exposed in rootstock cutting, the new root of the scion in the healing process will penetrate through the rootstock pith cavity into the soil, infecting scion seedlings with soil diseases; therefore, it is necessary to strictly control the age of rootstock seedlings. Acquiring the information of the structural characteristics of the pith cavity is an important pre-condition for precise cutting of rootstocks. The structural characteristics of the rootstock seedlings are shown in Figure 1. It can be obtained after analysis that the part from the pith cavity vertex to the base of the growing point is a solid structure, which is an important area of cutting surface of the rootstock. The cross-sections of melon rootstocks and scion seedling stems are both oval. The direction with the extension of cotyledons is the short axis, and that perpendicular to the short axis is defined as the long axis. It is required in splice grafting that there is a cutting surface along the direction of the short axis of the rootstock and scion seedling to ensure the connection of the vascular bundles on both cutting surfaces.

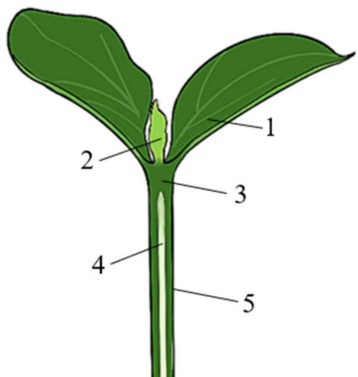

**Figure 1.** Pith cavity structure of the rootstock seedling: (1) cotyledon, (2) true leaf, (3) solid structure, (4) pith cavity, and (5) stem.

In order to better represent the internal features of the pith cavity of the rootstock seedling, a blade was used to uniformly cut along the spreading direction of the rootstock cotyledons, and a geometric model of the features of the rootstock pith cavity was constructed, as shown in Figure 2. It can be known after analysis that the intersection point $G$ of the two cotyledons is located between the pith cavity vertex, $O$, and the left growing point, $A_1$, and the right growing point, $A_2$. The geometric relationship between point $G$ and the internal pith cavity model was established to provide an external reference point for rootstock cutting. In Figure 2, the straight line $A_1E$ is the upper limit section after completely cutting a cotyledon and a growth point, and the straight line $A_1M$ is the lower limit section through the pith cavity vertex, $O$. Therefore, the cutting area of the rootstock was determined as $\Delta A_1ME$. In addition, trying to ensure the length of the cutting surface as long as possible, the thickness of the cutting surface, $F$, should also be considered. If $l_{op}$ is too small, there is a risk of cutting through the pith cavity, and it will have an adverse effect on the healing of the grafted seedlings. The test results showed that $l_{op} = 0.1–0.2$ mm could ensure that the grafted seedlings would have a high level of survival rate.

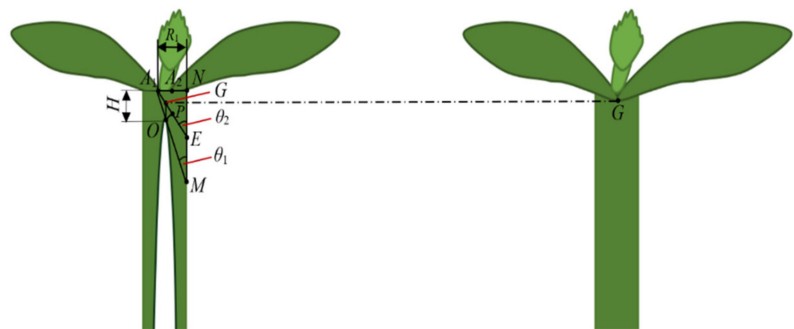

**Figure 2.** Rootstock cutting model: $A_1$ is the left base point of the rootstock growing point, $A_2$ is the right base point of the rootstock growing point, $O$ is the pith cavity vertex, and $G$ is the intersection point of two cotyledons.

The ultimate cutting angle is as follows:

$$\theta_1 = arc\tan\frac{R_2}{2H} \tag{1}$$

The ultimate cutting surface length is as follows:

$$l_{A_1M} = \frac{R_1}{\sin\theta_1} \tag{2}$$

The cutting angle through point $G$ is as follows:

$$\theta_2 = arc\tan\frac{R_2}{2(H - L_{OG})} \tag{3}$$

The cutting surface length through point $G$ is as follows:

$$l_{A_1E} = \frac{R_1}{\sin\theta_2} \tag{4}$$

The cutting surface thickness is as follows:

$$l_{OP} = \frac{R_2 sin(\theta_2 - \theta_1)}{2sin\theta_1} \tag{5}$$

where $R_1$ is the distance from the left base point of the growing point $A_1$ to the right edge of the seedling stem, mm; $R_2$ is the width of growing point, mm; $H$ is the height of the solid area, mm; and $L_{OG}$ is the vertical height from pith cavity vertex, $O$, to the point of intersection of two cotyledons $G$, mm.

The cutting angle of rootstock seedlings and the length and thickness of cutting surface can be obtained precisely and rapidly by analyzing the rootstock model. Then the data are used to match the cutting parameters of the scion seedling. The cutting model of the scion seedling is shown in Figure 3. The aim of grafting is to realize the same length of cutting surfaces of both rootstock and scion seedlings and improve the fitting precision and quality of grafted seedlings. Based on the cutting surface length of rootstock seedling, the cutting angle of the scion seedling can be calculated. The smaller the cutting angle, the longer the cutting surface and the thinner the cutting surface thickness; in this case, the scion seedling will be easily damaged when it is clamped by a clip.

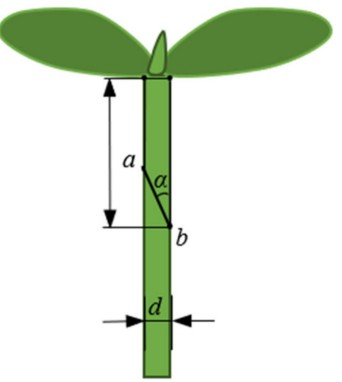

**Figure 3.** Cutting model of the scion seedling: $d$ is the length of the short axis of the scion seedling (mm), $l_{ab}$ is the length of cutting surface of the scion (mm), $\alpha$ is the cutting angle of the scion (°), and $h$ is the hypocotyledonary axis length after cutting the scion (mm).

The length of cutting surface of the scion is as follows:

$$l_{ab} = \frac{d}{sin\alpha} \tag{6}$$

If we place Equation (4) into Equation (6), we obtain the following.
The cutting angle of the scion is as follows:

$$\alpha = arcsin\frac{dsin\theta_2}{R_1} \tag{7}$$

where $d$ is the short axis of the scion stem, mm; $R_1$ is the distance from the left base point of the growing point of the rootstock to the right edge of seedling stem $N$, mm; and $\theta_2$ is the cutting angle on the rootstock (°).

Therefore, the cutting model by obtaining the characteristics information of the pith cavity in the rootstock and the point of intersection of two cotyledons can realize the precise cutting and grafting of rootstock and scion seedlings. By coupling the morphological information of the rootstock and pith cavity model through the visual image analysis system, the matched cutting information between rootstock and scion can be obtained; therefore, this grafting method can provide real-time information for intelligent cutting and improve the operation precision of the grafting machine.

*2.2. Visual Image Analysis System*

2.2.1. Structure of the System

The system includes a CCD camera, a PC and a holder, a data cable, and a seedling clamping module, as shown in Figure 4. The selected CCD camera is MV-EM120C (Shanxi Microvision Technology Co., Ltd., Xi'an, China), with 1.2 million effective pixels and a Gigabit Ethernet port to communicate with the PC in order to process and save the captured images; and the image storage format is .bmp, with a resolution of 1280 pixels × 960 pixels. The lens model is AFT-ZML1000, and the object distance range is 54–325 mm. The model of the image-processing software attached to the CCD camera is SVS-PLUS-4C, which has functional modules such as edge recognition, distance measurement, and feature positioning to provide abundant image-processing instructions. The operator can customize the operation interface and directly call the internal image-processing instructions of the system. Install the camera and the seedling clamping module on the fixing rod of the holder. The installation height can be adjusted freely. The installation distance between the camera and the seedling clamping module is 100–120 mm. Paste the buffer materials on the inside of the claw of the seedling clamping module to avoid damaging the seedlings.

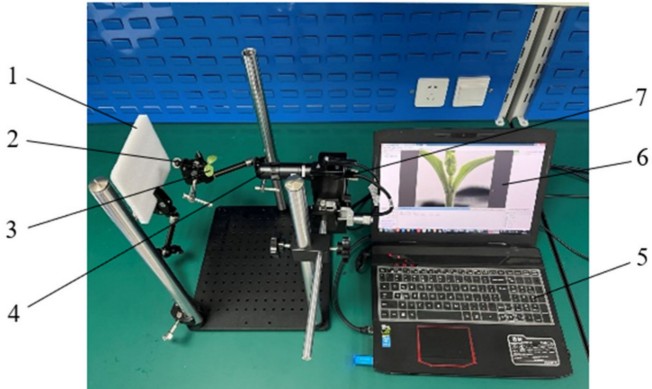

**Figure 4.** Structure of the visual system: (1) white board, (2) seedling clamping module, (3) seedling, (4) CCD camera, (5) PC, (6) software interface, and (7) holder.

### 2.2.2. Characteristic Identification and Extraction

The procedure of characteristic identification and extraction is as follows: (a) the camera collects clear images, (b) the image acquisition area is preprocessed; (c) the template is set, and the characteristic position in the image is calibrated; and (d) the image for contour recognition is then read, and then feature analysis and measurement are performed.

1. Gray processing

The shooting background has some influence on the quality of the image information collected by the camera, and a series of operations are required on the original image to highlight the characteristic parts for subsequent identification and measurement. The shooting background is white. The distribution ratio of the R–G–B value was adjusted by histogram grayscale transformation to adjust the color contrast of the seedlings and background. The histogram grayscale processing has linear and nonlinear function changes. The grayscale linear transformation is not conducive to highlighting the region of interest and is prone to cases of saturation and cutoff. In order to highlight the research object of interest in the image, it is necessary to enhance the contrast of a certain grayscale range. By using the method of piecewise linear stretch, the grayscale value of a certain range can be stretched, and the grayscale value of the other range can be compressed.

The equation for the piecewise linear stretch function is as follows:

$$f(x,y) = \begin{cases} (a/d)g(x,y)\ 0 \le g(x,y) < d \\ [(b-a)/(e-d)][g(x,y)-d]+a\ d \le g(x,y) < e \\ [(c-b)/f-e)][g(x,y)-e]+b\ e \le g(x,y) < f \\ \left[(M_g-b)/(M_f-c)\right][-f]+c\ f \le g(x,y) < M_g \end{cases} \tag{8}$$

where $g(x,y)$ is the read-in grayscale image, and $f(x,y)$ is the read-out grayscale image.

Compress and stretch the uninterested target interval, [*0*, *d*] and [*f*, *Mg*], and the interested target interval, [*d*, *f*], respectively, for the gray value of the image, and then obtain a clearer image.

The image of seedlings obtained without any external light sources was dark, so it was difficult to distinguish the contour features of seedlings, as shown in Figure 5a. After grayscale change, the color of the seedlings was darkened with clear contour lines, as shown in Figure 5b.

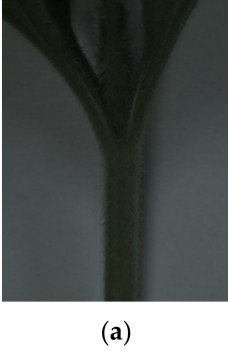
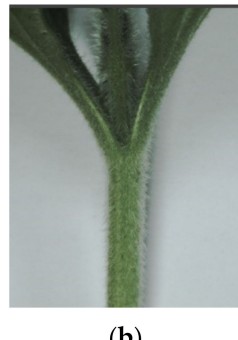

(**a**)  (**b**)

**Figure 5.** Grayscale change: (**a**) before grayscale change and, (**b**) after grayscale change.

2. Threshold segmentation

There are many fine hairs on the seedling stem and the growing point, and these hairs interfere with the extraction of contour lines, as shown in Figure 6a. The methods of noise mask and threshold segmentation were proposed to separate the seedling contour from the image background completely and clearly. By ignoring the mask module, the uninterested area is covered up. In real-time shooting, the uninterested area is ignored automatically, and the precise detection area is obtained, as shown in Figure 6b. At last, threshold segmentation was applied to reduce the noise interference of the hair edge. The calculation results showed that the range of threshold segmentation was 128–170, showing an obvious segmentation effect, and the contour features of seedlings can be clearly identified after this processing, as shown in Figure 6c.

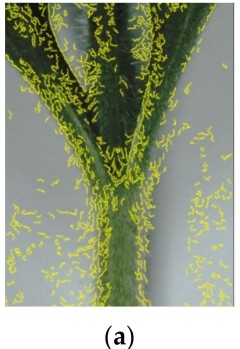
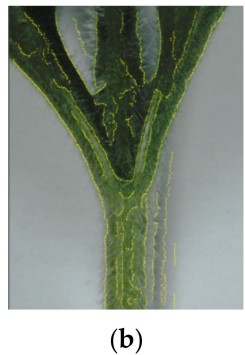
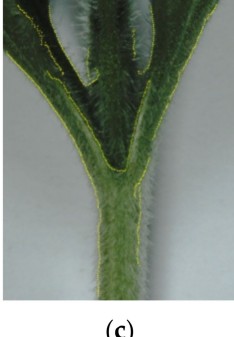

(**a**)  (**b**)  (**c**)

**Figure 6.** Image feature extraction: (**a**) initial state, (**b**) mask processing, and (**c**) threshold segmentation.

3. Feature measurement

The data obtained after feature measurement are the real-time pixel values; therefore, it is necessary to calibrate the measured pixel values by a measuring scale. First, a camera was used to shoot and measure a standard ruler. In the interface of scale calibration in the visual software, set the datum length as actual length and click "apply" to finish scale conversion and calibration. Since the pixel values of an image are associated with object distance of the camera, scale calibration should be performed on pixel values before each time of measurement. After template feature extraction on the measuring position, based on the fact that the edge features of the seedlings show irregular linear distribution, the outer contour feature line (discrete type) obtained by edge detection undergoes curve fitting with the edge of the features to be measured. Then measure the vertical distance of both sides of the curve after fitting, and the software system automatically calculates the mean value, as shown in Figure 7a,b. Feature location is carried out on the point of intersection of two rootstock cotyledons to make the contour lines of them. Take the point of intersection of the two contour lines as that of the rootstock cotyledons, and then save the template, as shown in Figure 7c.

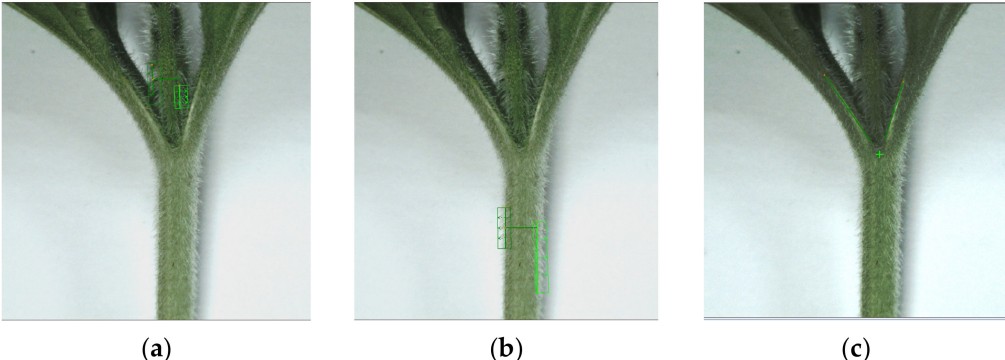

(a)        (b)        (c)

**Figure 7.** Measuring the morphological features of the seedling: (**a**) measuring features of the growing point, (**b**) measuring features of seedling stem, and (**c**) calibrating intersection features of cotyledons.

*2.3. Test Method*

2.3.1. Test Materials

Jingxinzhen No. 4 *Cucurbita moschata* and Jingmei watermelon were chosen as the rootstock and the scion, respectively, in the test, which was carried out by Jingyan Yinong (Beijing) Seed Sci-Tech Co., Ltd., Beijing China, in the greenhouse of Beijing Academy of Agriculture and Forestry, from 14 to 28 November 2021 (Geographic coordinates: 116.2° E, 39.9° N). The seedling cultivation process is as follows: first, soak the rootstock and scion seeds in warm water at 55 °C and stir for 3 h; and then place the seeds in a BSC-800 box, with constant temperature and humidity (Shanghai Boxun Industrial Co., Ltd., Shanghai China), for pre-germination for 24 h at the temperature of 30 °C and a humidity of 90%. The substrate for seedlings is vermiculite, peat, and perlite mixed and stirred at a ratio of 1:1:1, and 50-hole and 72-hole sowing trays were used for the rootstock and scion, respectively. After sowing, the seedlings were placed into the greenhouse for cultivation, and the environmental condition was as follows: the temperature at 25–28 °C, humidity of 60–80%, and seedling cultivation time for 12–14 days. After the rootstock grew to two leaves and one core, and the scion grew to two unfolded cotyledons, the measurement test was carried out, as shown in Figure 8.

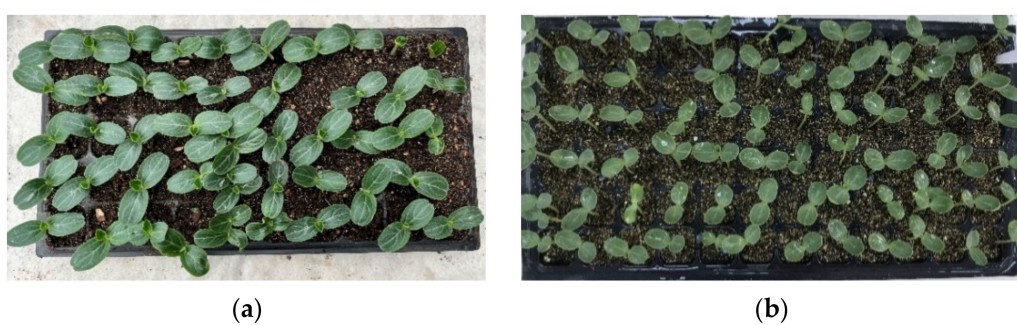

(a)        (b)

**Figure 8.** Test materials: (**a**) rootstock seedlings in a greenhouse and, (**b**) scion seedlings in a greenhouse.

2.3.2. Morphological Parameters of Seedlings

The morphological parameters of seedlings were measured, as shown in Figure 9. Cut the rootstock and scion seedling stems along the substrate surface and prepare 50 test samples. The test process is as follows: place seedlings (rootstock and scion seedlings) into the seedling clamping module, let the cotyledon unfolding direction perpendicular to the shooting direction of the camera, and adjust the microscope lens to clarify the image of the seedling. Set the parameter values, such as grayscale processing and threshold segmentation; make scale calibration on the measured data by the camera; set the edge lines of long and short axis of the seedling stem, edge lines on both sides of the growing point, edge lines of the cotyledons, the reference point of cotyledon root, and reference

line of cutting surface of stems; establish and save the measurement template; make real-time analysis with the visual software system; and calculate the geometric parameters of morphological features of the seedlings.

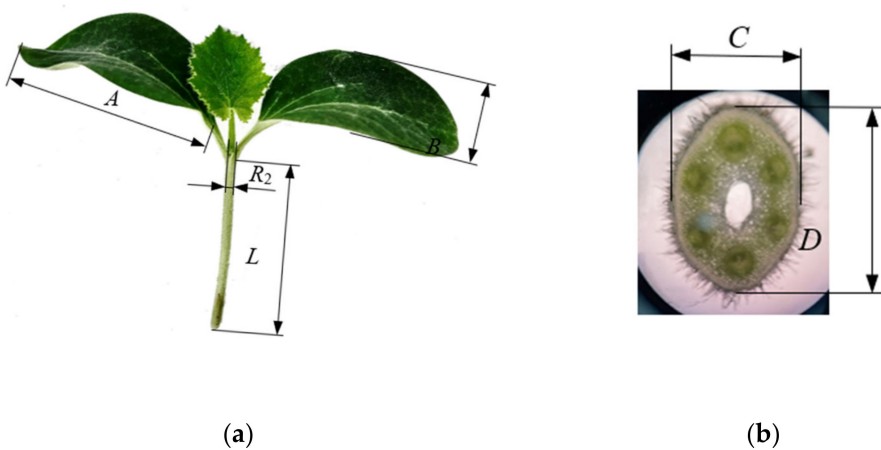

(**a**)    (**b**)

**Figure 9.** Morphological parameters of the seedlings: (**a**) rootstock seedling and, (**b**) cross-section of the stem. *L* is seedling height, *A* is cotyledon length, *B* is cotyledon width, *C* is long axis of the stem, *D* is short axis of the stem, and $R_2$ is the width of the growing point.

### 2.3.3. Cutting Model of the Rootstock Seedling

Split the rootstock seedling uniformly along the unfolding direction of the cotyledons and place the seedling in the image acquisition area of the camera. According to the edit instruction of the visual image analysis software, calibrate the vertex of the pith cavity, *O*, in the image, left point $A_1$ and right point $A_2$ of the growing point, and determine the cutting area of the rootstock, which is the solid structure above the vertex of the pith cavity. Then, taking the cross-section of the stem as the basis for reference, fetch the position information of the point of intersection of cotyledons, *G*; analyze and establish the geometric analysis model of the pith cavity; and the point of intersection of the two cotyledons, as shown in Figure 10. In this way, by taking the point of intersection of two cotyledons as the reference point of the cutting operation, the cutting angle and area are precisely measured to match the cutting parameters of the scion seedling. Then prepare 50 sample seedlings for the test.

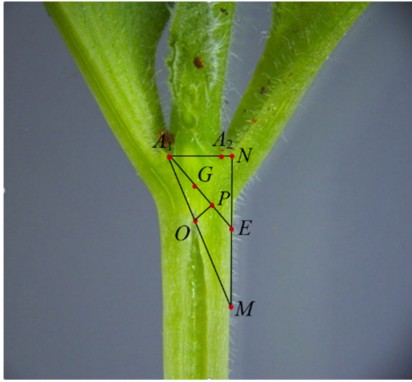

**Figure 10.** Cutting model of the rootstock seedling.

## 3. Results and Discussion

The statistical results of the morphological parameters of rootstock and scion seedlings are shown in Table 1.

**Table 1.** Morphological parameters of seedlings.

| Object | Seedling Height L (mm) | Seedling Stem Long Axis D (mm) | Seedling Stem Short Axis C (mm) | Length–Width Ratio of Seedling Stem $\zeta$ | Cotyledon Length A (mm) | Cotyledon Width B (mm) | Growing Point Width $R_2$ (mm) |
|---|---|---|---|---|---|---|---|
| Rootstock seedling | 43.57 ± 2.65 (CV 6.08%) | 2.41 ± 0.18 (CV 7.46%) | 2.23 ± 0.16 (CV 7.17%) | 1.08 | 37.60 ± 3.14 (CV 8.35%) | 25.44 ± 2.56 (CV 10.06%) | 1.21 ± 0.10 (CV 8.26%) |
| Scion seedling | 34.42 ± 3.21 (CV 9.33%) | 2.34 ± 0.19 (CV 8.12%) | 1.66 ± 0.13 (CV 7.83%) | 1.41 | 22.79 ± 2.84 (CV 12.46%) | 16.76 ± 0.88 (CV 5.25%) | - |

Note: 50 rootstock seedlings and 50 scion seedlings were prepared as sample seedlings; ± values mean standard deviation.

Table 1 shows that the rootstock seedling height is 43.57 ± 2.65 mm; the long axis and short axis of the seedling stem are 2.41 ± 0.18 mm and 2.23 ± 0.16 mm, respectively; the cotyledon length and width are 37.60 ± 3.14 mm and 25.44 ± 2.56 m, respectively; the growing point width is 1.21 ± 0.1 mm; the plant height of scion seedlings is 34.42 ± 3.21 mm; the long axis and short axis of seedling stem are 2.34 ± 0.19 mm and 1.66 ± 0.13 mm, respectively; and the cotyledon length and width are 22.79 ± 2.84 mm and 16.76 ± 0.88 mm, respectively. Since the rootstock and scion were sown during the same day, their seedling age was the same, and the rootstock seed was small grain of *Cucurbita moschata*; they had little difference in morphological features. In addition, the length–width ratios of the rootstock and scion were 1.08 and 1.41, respectively, indicating that the length and short axes of the rootstock seedling stems were not much different, and the scion seedling stems were relatively flat. The difference in long axes of rootstock and scion seedlings was not much, with a difference of only 0.18 mm, and the short axis of the rootstock is 0.57 mm longer than that of the scion. Therefore, the grafting accuracy cannot be guaranteed by adopting a fixed cutting angle, showing that it is necessary to match the rootstock and the scion seedlings in grafting. In order to match the rootstock and scion seedlings in grafting, it is necessary to further determine the structural parameters of the pith cavity in the rootstock, as well as the position information of the point of intersection of rootstock cotyledons, to finally determine the cutting model of the rootstock seedling.

The statistical results of the structural parameters of the pith cavity of the rootstock seedling are shown in Table 2.

**Table 2.** Structural parameters of pith cavity of rootstock seedlings.

| Object | Left Base Point of Growing Point to the Right Edge of Seedling Stem, $R_1$ (mm) | Width of the Growing Point, $R_2$ (mm) | Stem Height, $H$ (mm) | Distance from Pith Cavity Vertex to the Point of Intersection of Cotyledons, $L_{OG}$ (mm) |
|---|---|---|---|---|
| Rootstock seedling | 1.88 ± 0.15 (CV 7.97%) | 1.21 ± 0.10 (CV 8.26%) | 1.84 ± 0.12 (CV 6.52%) | 0.79 ± 0.16 (CV 20.22%) |

Note: 50 rootstock seedlings and 50 scion seedlings were prepared as sample seedlings; ± values mean standard deviation.

Table 2 shows that the distance from the left base point of the growing point to the right edge of seedling stem $R_1$ is 1.88 ± 0.15 mm; $R_1$ and the short axis of scion seedling stem $d$ are the important parameters in matching scion and rootstock in grafting. The height of the seedling stem, $H$, is 1.84 ± 0.12 mm, indicating that the rootstock cutting is very difficult and will easily cut off the pith cavity; thus, it is necessary to control the cutting angle precisely. The distance from pith cavity vertex to the point of intersection of cotyledons, $L_{OG}$, was 0.79 ± 0.16 mm. Considering the position information of left base point $A_1$ of the growing point, the trajectory of cutting one piece of cotyledon of the rootstock and the growing point was determined to be a straight line, $A_1E$, and the trajectory of cutting through the pith cavity vertex, $O$, was a straight line, $A_1M$; thus, the cutting area on the rootstock was determined to be $\triangle A_1ME$ (Figure 10). In order to further determine the cutting angle for optimal matching effect of rootstock and scion, the data in Tables 1 and 2 are imported into Equations (2) and (4)–(6). The ultimate cutting parameters

of rootstock seedlings were obtained, and then we matched the cutting angle through point *G* with the cutting angle of the scion; the results are shown in Table 3.

**Table 3.** Ultimate cutting parameters of the rootstock.

| Object | Parameters | Values |
|---|---|---|
| Rootstock seedling | The ultimate cutting angle, $\theta_1$ (°) | $18.21 \pm 1.92$ |
| | The ultimate cutting surface length, $l_{A_1M}$ (mm) | $6.00 \pm 0.76$ |
| | The cutting angle through point $G$, $\theta_2$ (°) | $28.32 \pm 2.56$ |
| | The cutting surface length through point $G$, $l_{A_1E}$ (mm) | $3.95 \pm 0.65$ |
| | The cutting surface thickness through point $G$, $l_{op}$ (mm) | $0.31 \pm 0.08$ |
| Scion seedling | Cutting angle, $\alpha$ (°) | $24.71 \pm 2.83$ |

Table 3. shows that, the ultimate cutting angle of the rootstock $\theta_1$ and ultimate cutting surface length $l_{A_1M}$ are $18.21 \pm 1.92°$ and $6.00 \pm 0.76$ mm, respectively, which can ensure the safety limit by avoiding cutting off the pith cavity in cutting the rootstock. Besides, the cutting angle $\theta_2$ through point G, cutting surface length $l_{A_1E}$ and cutting surface thickness $l_{op}$ are $28.32 \pm 2.56°$, $3.954 \pm 0.65$ mm and $0.31 \pm 0.08$ mm, respectively, which are the minimum limit value to ensure cutting exactly one piece of cotyledon and the growing point, and under such condition, the cutting angle of the scion $\alpha$ is $24.71 \pm 2.83°$. Taking exactly cutting off one piece of cotyledon and the growing point as the target, then the rootstock cutting surface length $l_{A_1E}$ is $3.954 \pm 0.65$ mm. If the rootstock cutting surface is too short, the survival rate of grafted seedlings after recovery would be affected. Considering that the cutting surface thickness $l_{op}$ is $0.31 \pm 0.08$ mm, the cutting surface length can be increased by slightly reducing the cutting surface thickness. For this purpose, method of interpolation was used to calculate the rootstock cutting surface length and thickness at each interval of 2° within the range of cutting angle 18.21–28.32° (cutting angles of 20°, 22°, 24°and 26°), to match the cutting angles of the scion seedlings, the matching results are shown in Table 4.

**Table 4.** Matching results of rootstock and scion after cutting.

| Cutting Angle of the Rootstock (°) | Length of Cutting Surface Length (mm) | Thickness of Cutting Surface Length (mm) | Cutting Angle by Matching Rootstock and Scion (°) | Vertical Height of the Cutting Surface of the Scion (mm) |
|---|---|---|---|---|
| 20 | $5.43 \pm 0.44$ | $0.06 \pm 0.01$ | 17.91 | $5.17 \pm 0.47$ |
| 22 | $4.96 \pm 0.40$ | $0.13 \pm 0.02$ | 19.68 | $5.12 \pm 0.48$ |
| 24 | $4.57 \pm 0.37$ | $0.20 \pm 0.01$ | 21.45 | $5.06 \pm 0.48$ |
| 26 | $4.24 \pm 0.35$ | $0.27 \pm 0.02$ | 23.22 | $4.99 \pm 0.49$ |

Table 4 shows that, when the cutting angle of the rootstock is 20°, the cutting surface length and thickness are $5.43 \pm 0.44$ mm and $0.06 \pm 0.01$ mm, respectively. When the cutting surface thickness is lower than 0.1 mm, there is a very high risk of cutting through the pith cavity; thus, the safety in cutting cannot be ensured. On the premise of safety cutting of the rootstock, it is necessary to follow the principle that, the longer the cutting surface is, the better it is for the recovery of the grafted seedling. When the cutting angle of the rootstock was 22–26°, the cutting surface thickness was 0.13–0.27 mm, and the cutting surface length was 4.96–4.24 mm. All parameters could meet the requirements of the melon grafting operation; the operation accuracy of the existing cutting mechanism can reach ±0.01 mm; therefore, the rootstock cutting angle could be determined as 22°, and the corresponding cutting surface length and thickness were 4.96 mm and 0.13 mm, respectively. Compared with the cutting surface length, $l_{A_1E}$, through point *G* in Table 3, the length increased by 1.01 mm, indicating that the result was more conducive to the healing and survival of grafted seedlings. Using the grafting model constructed in this paper for calculation, we see that the corresponding cutting angle of the scion was 19.68°, and the vertical height of the scion cutting surface was $5.12 \pm 0.47$ mm, which could also meet the requirements in clamping and fixing the grafting clip for the grafted seedlings (the height of the clipping mouth of the grafting clip is 10 mm).

In manual grafting, hand–eye coordination is required to fit the cutting surfaces of the rootstock and the scion together. In order to facilitate the matching operation, the

cutting surface length of the scion should be generally slightly larger than that of the cutting surface of the rootstock. After the fitting of the two cutting surfaces, a part of the cutting surface of the scion was exposed, as shown in Figure 11a. During the machine grafting operation, by cutting at a fixed cutting angle, the cutting surface length of the scion was slightly less than that of the rootstock, and after the two cutting surfaces fit each other, a part of the cutting surface of the rootstock was partially exposed, as shown in Figure 11b. Therefore, manual- and machine-grafted seedlings are at risk of infection, and it is necessary to perform disinfection treatment after grafting in a timely manner. By analyzing the structural characteristics of the pith cavity of the rootstock, a cutting model matching the rootstock and the scion was constructed, and the consistency of the length of the rootstock and scion cutting surfaces was achieved, as shown in Figure 11c. The grafting method improved the safety of rootstock grafting operation and avoided cutting through the pith cavity. Compared with manual and machine grafting, the grafting method can greatly improve the accuracy of matching precision of grafted seedlings, and this helps to improve the survival rate of grafted seedlings.

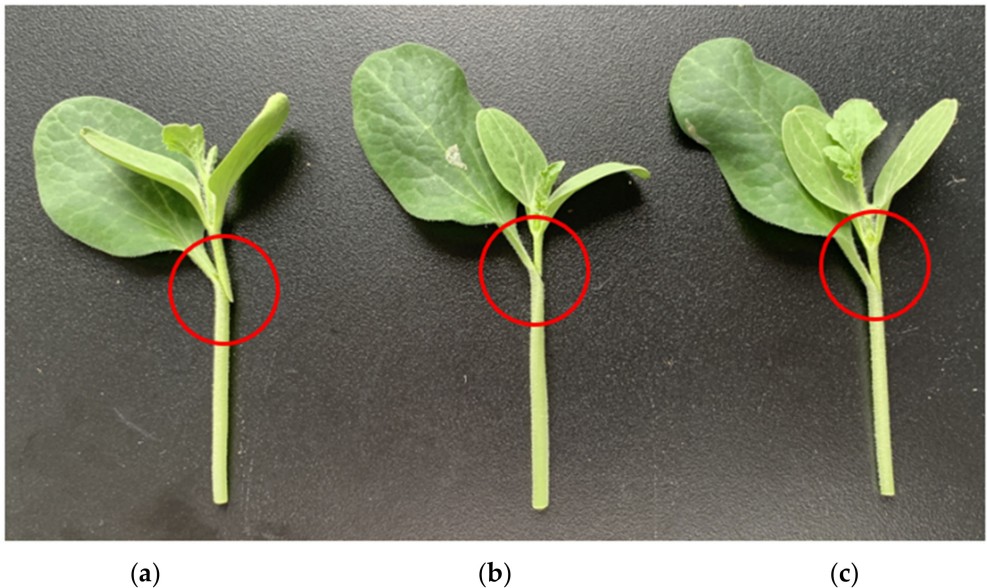

(**a**)　　　　　　　　　　(**b**)　　　　　　　　　　(**c**)

**Figure 11.** Fitting-precision comparison of different grafting methods: (**a**) manual grafting, (**b**) machine grafting, and (**c**) matched grafting.

The splice grafting for melons based on visual image was proposed in this paper, and the consistency of cutting surfaces of rootstock and scion seedlings was achieved; thus, it is a key technology in improving the cutting precision of seedlings and grafting quality. If this technology can be applied to the existing grafting machine, it will help alternate the operation mode of adjusting the parameters of the cutting mechanism based on operator's personal experience, and the morphological information of each group of grafted seedlings could be obtained in real time to provide automatic guidance for adjusting the parameters of the cutting mechanism. The research results can provide an important reference for the optimal design and intelligent upgrade of the grafting machine cutting mechanism and will become an important trend in the development of intelligent grafting technology and equipment in the future.

## 4. Conclusions

(1)　Based on the existing grafting machine, the operating parameters of the cutting mechanism need to be adjusted by manual experience, and the cutting angle of the seedlings cannot be obtained, resulting in problems such as the low fitting accuracy of the grafted seedlings and low survival rate. In this paper, a visual image-based grafting method for melons was proposed. The visual system was used to obtain

    the structure of the pith cavity and morphological information of the rootstock, and the grafting model of the rootstock and the scion were constructed to realize the standardized and precise cutting operation of them.

(2)   In order to obtain the geometric model of the cutting area of the rootstock, a visual image analysis system was established to identify, analyze, and model the pith cavity structure inside the rootstock seedlings and morphological characteristics, and the ultimate cutting angle of the rootstock seedlings and cutting surface parameters were determined. By measuring the length of the short axis of scion seedlings to achieve the optimal fitting of them, the optimal cutting angle of the rootstocks and scion seedlings was determined, and then grafting and seedling cultivation tests were carried out.

(3)   The test results showed that the cutting angle on rootstock seedlings was $18.21 \pm 1.92°$, and the matching cutting angles of the rootstock and scion seedlings were $22°$ and $19.68°$, respectively. The cutting surface length was 4.96 mm, and cutting surface thickness of the rootstock was 0.13mm, which could satisfy the technological requirements in splice grafting of melons. The research results realized the concept of maximizing the cutting surfaces of the rootstock and the scion, which can be applied to the technology development, optimization, and upgrading of the cutting mechanism of the grafting machine.

## 5. Patents

    Kai Jiang, Wenzhong Guo, Xiaoming Wei, Dongdong Jia, and Wenqian Huang. Rootstock–Scion Matched Grafting Method, Apparatus, and Grafting Robot for Melon Seedling. Application Number: ZL 202111656817.0. China National Intellectual Property Administration.

**Author Contributions:** Conceptualization, P.X., T.Z. and K.J.; methodology, P.X., T.Z. and K.J.; software, L.C. and W.H.; writing—original draft preparation, P.X., T.Z. and K.J.; writing—review and editing, L.C. and K.J.; visualization, T.Z. and P.X.; supervision, L.C. and K.J.; funding acquisition, K.J. All authors have read and agreed to the published version of the manuscript.

**Funding:** This research was funded by the National key Research and Development Program of China (2019YFE0125200), the National Nature Science Foundation of China (Grant No. 32171898), the BAAFS Innovation Ability Project (KJCX20220403), and the China National Agricultural Research System (CARS-25-07).

**Institutional Review Board Statement:** Not applicable.

**Informed Consent Statement:** Not applicable.

**Data Availability Statement:** The data presented in this study are available upon demand from the correspondence author at (jiangk@nercita.org.cn).

**Conflicts of Interest:** The authors declare no conflict of interest.

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
