# Peer review of "Study on the Method of Matched Splice Grafting for Melon Seedlings Based on Visual Image"

_agriculture, doi:10.3390/agriculture12070929_

Round 1

Reviewer 1 Report

This manuscript describes the grafting design for melon rootstock seedling by using visual images. This manuscript may be helpful for future research on the cutting mechanism of the grafting machine. There are major problems that should be corrected in the revised version. 

Lines 45-50: Authors should add the references of each of these sentences.

Lines 58-65: Authors should add the references of each of these sentences.

Lines 130-143: Authors should add the references of each of these sentences.

Figure 11: Authors should change the drawing pictures to the real pictures for identification of three grafting methods in real life. Although the manuscript described well on the grafting methods, the authors did not provide the real pictures of the machined techniques, and the survival seedlings which identified the grafting points.

Finally, the research was not conducted correctly because the authors did not show the real pictures of new grafting methods.

Author Response

Response to Reviewer 1 Comments

Dear Editors and Reviewers,

On behalf of all the authors, I would like to sincerely appreciate your valuable comments on the manuscript. Your comments not only provide constructive suggestions on improving the quality of the manuscript, but also lead us to in-depth thinking of our approaches. We will benefit from them for our future research. Based on your review comments, we have revised the manuscript accordingly and highlighted the changes. In the following, we described the changes we made corresponding to each comment.

Point 1:

1) Lines 45-50: Authors should add the references of each of these sentences.

2) Lines 58-65: Authors should add the references of each of these sentences.

3) Lines 130-143: Authors should add the references of each of these sentences.

4) Figure 11: Authors should change the drawing pictures to the real pictures for identification of three grafting methods in real life. Although the manuscript described well on the grafting methods, the authors did not provide the real pictures of the machined techniques, and the survival seedlings which identified the grafting points.

5) Finally, the research was not conducted correctly because the authors did not show the real pictures of new grafting methods.

Authors’ Response: We really appreciate your positive and constructive comments on our manuscript. The manuscript was revised carefully based on the comments.

 Response 1:

1) Lines 45-50: Authors should add the references of each of these sentences.

Authors’ Response: We have added references in line 49.

2) Lines 58-65: Authors should add the references of each of these sentences.

Authors’ Response: We have added references in lines 60 and 64.

3) Lines 130-143: Authors should add the references of each of these sentences.

Authors’ Response: We have added references in lines 141, 147 and 152.

4) Figure 11: Authors should change the drawing pictures to the real pictures for identification of three grafting methods in real life. Although the manuscript described well on the grafting methods, the authors did not provide the real pictures of the machined techniques, and the survival seedlings which identified the grafting points.

Authors’ Response: We have replaced the original image with a real image, please see Fig. 11. The innovation of this manuscript lies in the method of matched splice grafting for melon seedlings that is applied in mechanized grafting, and the related study on the principles, realization process and tests of this method. In this research, the authors did not specify the operation of the guiding machinery and seedling survival rate, which will be further studied and described in details in our future manuscript.

5) Finally, the research was not conducted correctly because the authors did not show the real pictures of new grafting methods.

Authors’ Response: The pictures below show the grafting comparison test carried out by our research group, including matched grafting, manual grafting and machine grafting, and test results are being sorted.

Reviewer 2 Report

1.   Abstract is still too wordy. Please, avoid repetition of sentences. Authors cite parameters that are not introduced before and it is hard to distinguish the literature review from their contribution.

2.   Introduction section still needs to be improved. Authors do not present properly the context or the issue that they are trying to solve with their contribution. Literature review is not properly ordered and it is still difficult to read. Additionally, other sections in the manuscript still include information that should be presented in the Introduction as literature review.

3.  Much information that is not relevant is included in the text. However, the real contribution of the paper is not thoughtfully explained.

4.  The model proposed is still not clearly explained. Authors jump from one step to the next with no clear explanation of the aim of each transformation or calculation the accomplish. There is still no general view of the purpose of the model.

5.    Quality of some figures needs to be improved.

Author Response

Response to Reviewer 2 Comments

Dear Editors and Reviewers,

On behalf of all the authors, I would like to sincerely appreciate your valuable comments on the manuscript. Your comments not only provide constructive suggestions on improving the quality of the manuscript, but also lead us to in-depth thinking of our approaches. We will benefit from them for our future research. Based on your review comments, we have revised the manuscript accordingly and highlighted the changes. In the following, we described the changes we made corresponding to each comment.

Point 2:

1) Abstract is still too wordy. Please, avoid repetition of sentences. Authors cite parameters that are not introduced before and it is hard to distinguish the literature review from their contribution.

2) Introduction section still needs to be improved. Authors do not present properly the context or the issue that they are trying to solve with their contribution. Literature review is not properly ordered and it is still difficult to read. Additionally, other sections in the manuscript still include information that should be presented in the Introduction as literature review.

3) Much information that is not relevant is included in the text. However, the real contribution of the paper is not thoughtfully explained.

4) The model proposed is still not clearly explained. Authors jump from one step to the next with no clear explanation of the aim of each transformation or calculation the accomplish. There is still no general view of the purpose of the model.

5) Quality of some figures needs to be improved.

Authors’ Response: We really appreciate your positive and constructive comments on our manuscript. The manuscript was revised carefully based on the comments.

 Response 2:

1) Abstract is still too wordy. Please, avoid repetition of sentences. Authors cite parameters that are not introduced before and it is hard to distinguish the literature review from their contribution.

Authors’ Response: We have shortened the abstract to make it brief in lines 12-15. In the Introduction, we have introduced in details the structural characteristics of current melon grafting machine and its operation modes, and analyzed the key problems on melon grafting machine that have not been tackled yet, then we proposed the solution to the problems above in this study.

 2)  Introduction section still needs to be improved. Authors do not present properly the context or the issue that they are trying to solve with their contribution. Literature review is not properly ordered and it is still difficult to read. Additionally, other sections in the manuscript still include information that should be presented in the Introduction as literature review.

Authors’ Response: The manuscript proposed a matched grafting method based on visual image. Therefore, in the Introduction section, we added the content in previous studies on detection of seedling’s morphological characteristics based on machine vision technology in lines 107-104, we also described the current studies on melon grafting machine at present, and proposed that, it was impossible to realize machine’s adaptivity to seedlings based on a fixed angle in seedling cutting. Therefore, the authors proposed a matched grafting method that acquires the morphological information of seedlings through visual technology, to realize the real-time acquisition of parameters in matched grafting. The method can guide the cutting mechanism to adjust the cutting angle, realize the consistent incision length of the rootstock and the scion, to ensure the precision grafting of the machine.

 3)  Much information that is not relevant is included in the text. However, the real contribution of the paper is not thoughtfully explained.

Authors’ Response:  The aim of this study is to solve the problem of failure in offering a matched cutting angle of the grafting machine. First, the matched grafting model between the rootstock and the scion was established and the cutting operation parameters of the seedlings were determined. Then a visual test platform was constructed to obtain and recognize the morphological characteristics of the seedlings and the pith cavity structure, and related data on matched grafted. At last, the manuscript discussed the test results, and compared the advantages and disadvantages of manual grafting and machine-group, and proposed a guiding cutting mechanism based on visual image information. To provide accurate basis for the accurate operation of grafting robots.

4)  The model proposed is still not clearly explained. Authors jump from one step to the next with no clear explanation of the aim of each transformation or calculation the accomplish. There is still no general view of the purpose of the model.

Authors’ Response: In section 2.1, we proposed the matched grafting method and calculation model, to keep the consistency in length of the rootstock and the scion. In section 2.2, we proposed the method of obtaining morphological information of seedlings based on machine vision technology, and specified the process of analyzing the visual system. In section 2.3, we tested the structural parameters of the morphological characteristics of the seedling and the pith cavity of the rootstock. In section 3, we discussed the data of cutting parameters for matched grafting of the rootstock and the scion. The contents above were performed according to the steps of proposing a theoretical model, technological realization and test analysis with explicit targets for each step. Thus, the test results can provide reference for accurate grafting. At present, we are working on the comparative test research on the three grafting methods and are sorting and analyzing the results.

5) Quality of some figures needs to be improved.

Authors’ Response: Reply: Thank you for your valuable comments. We have re-checked the figures and made revisions.

Reviewer 3 Report

Dear colleagues! The authors' research is of applied interest for improving the technology of grafting melons.
The authors described in detail the method of manipulations with plant seedlings, provided detailed illustrations with geometric characteristics.
I think that there are minor shortcomings in the manuscript in the statistical aspect. There are some recommendations and questions for the authors. 1. In tables 1-3 and the text after them, the bit depth of three decimal places should be reduced to the average statistical deviation. 2. Tables 1-2 provide data on morphometric indicators of seedlings.
In order to find out whether the sample is homogeneous, it is necessary to conduct a statistical analysis for the presence of individual variability of plant seedlings.
3. It is necessary to conduct a statistical analysis of the data compared in Table 4 for the reliability of the differences.

Author Response

Response to Reviewer 3 Comments

Dear Editors and Reviewers,

On behalf of all the authors, I would like to sincerely appreciate your valuable comments on the manuscript. Your comments not only provide constructive suggestions on improving the quality of the manuscript, but also lead us to in-depth thinking of our approaches. We will benefit from them for our future research. Based on your review comments, we have revised the manuscript accordingly and highlighted the changes. In the following, we described the changes we made corresponding to each comment.

Point 3:

1) In tables 1-3 and the text after them, the bit depth of three decimal places should be reduced to the average statistical deviation. 

2) Tables 1-2 provide data on morphometric indicators of seedlings. In order to find out whether the sample is homogeneous, it is necessary to conduct a statistical analysis for the presence of individual variability of plant seedlings.

3) It is necessary to conduct a statistical analysis of the data compared in Table 4 for the reliability of the differences.

Authors’ Response: We really appreciate your positive and constructive comments on our manuscript. The manuscript was revised carefully based on the comments.

 Response 3:

1) In tables 1-3 and the text after them, the bit depth of three decimal places should be reduced to the average statistical deviation. 

Authors’ Response: In tables 1-3 and the text after them, the bit depth had two decimal places. So we don’t quite understand the comment “the bit depth of three decimal places should be reduced to the average statistical deviation”, if there is misunderstanding, please feel free to tell us the problem.

2) Tables 1-2 provide data on morphometric indicators of seedlings. In order to find out whether the sample is homogeneous, it is necessary to conduct a statistical analysis for the presence of individual variability of plant seedlings.

Authors’ Response: We have added the coefficient of variation in the morphological parameters of seedlings in Tables 1-2.

3) It is necessary to conduct a statistical analysis of the data compared in Table 4 for the reliability of the differences.

Authors’ Response: Table 4 was obtained after further analyzing the cutting parameters of the rootstock in Table 3. Within the range of ultimate cutting angle of the rootstock, four groups of cutting data were obtained, and the incision length and incision thickness were taken as the limiting conditions, finally the cutting data for matched grafting between the rootstock and the scion were obtained. Therefore, there is no need to do statistical analysis on Table 4.

Round 2

Reviewer 1 Report

The manuscript was revised carefully. It is more interesting than the previous one. It was rejected the first time; however, the Reviewer considers the second submission to be good. So, the Reviewer accepts it in the present form.